# Global dynamics of neural mass models

**Gerald Kaushallye Cooray**[1,2,3]*, **Richard Ewald Rosch**[2,4,5], **Karl John Friston**[4]

**1** GOS-UCL Institute of Child Health, University College London, London, United Kingdom, **2** Great Ormond Street Hospital NHS Foundation Trust, London, United Kingdom, **3** Karolinska Institutet, Stockholm, Sweden, **4** The Wellcome Centre for Human Neuroimaging, Queen Square Institute of Neurology, University College London, London, United Kingdom, **5** MRC Centre for Neurodevelopmental Disorders, Institute of Psychiatry, Psychology and Neuroscience, King's College London, United Kingdom

* gerald.cooray@ki.se

**Data Availability Statement:** All codes required to repeat the computations are available at https://github.com/gercoo/exittimes2synapticconnectivity. The SPM12 toolbox (https://www.fil.ion.ucl.ac.uk/spm/software/spm12/) is required.

## Abstract

Neural mass models are used to simulate cortical dynamics and to explain the electrical and magnetic fields measured using electro- and magnetoencephalography. Simulations evince a complex phase-space structure for these kinds of models; including stationary points and limit cycles and the possibility for bifurcations and transitions among different modes of activity. This complexity allows neural mass models to describe the itinerant features of brain dynamics. However, expressive, nonlinear neural mass models are often difficult to fit to empirical data without additional simplifying assumptions: e.g., that the system can be modelled as linear perturbations around a fixed point. In this study we offer a mathematical analysis of neural mass models, specifically the canonical microcircuit model, providing analytical solutions describing slow changes in the type of cortical activity, i.e. dynamical itinerancy. We derive a perturbation analysis up to second order of the phase flow, together with adiabatic approximations. This allows us to describe amplitude modulations in a relatively simple mathematical format providing analytic proof-of-principle for the existence of semi-stable states of cortical dynamics at the scale of a cortical column. This work allows for model inversion of neural mass models, not only around fixed points, but over regions of phase space that encompass transitions among semi or multi-stable states of oscillatory activity. Crucially, these theoretical results speak to model inversion in the context of multiple semi-stable brain states, such as the transition between interictal, pre-ictal and ictal activity in epilepsy.

## Author summary

There is collecting evidence that the electrical activity of the brain is highly complex. Electro- and magnetoencephalography, being non-invasive methods, are often used to measure brain function in humans and have provided a large amount of data indicating complexity. Furthermore, a quantitative framework is required to understand the interactions between the different features of electrical activity of the brain. Neural mass models can provide a quantitative framework required to further understand brain activity and complexity, and in this paper we provide a mathematical form that can be used to understand some of the complex electrical dynamics of the brain. Crucially, these theoretical

**Funding:** KJF is supported by funding for the Wellcome Centre for Human Neuroimaging (Grant number: 205103/Z/16/Z) and a Canada-UK Artificial Intelligence Initiative (Grant number: ES/T01279X/1). RER is supported by funding from the Wellcome Trust (Grant number: 209164/Z/17/Z). KJF and RER are supported by the European Union's Horizon 2020 Framework Programme for Research and Innovation, Human Brain Project SGA3 (Grant Number: 945539). The funders had no role in study design, data collection and analysis, decision to publish, or preparation of the manuscript.

**Competing interests:** There are no competing interests for any of the authors.

results improve our understanding of multiple semi-stable brain states, as seen in epilepsy.

This is a *PLOS Computational Biology* Methods paper.

## Introduction

The surface of the human brain is covered by a cortical layer of grey matter. This cortex contains histologically distinguishable layers with laminar-specific types of neural cells that show patterned interlaminar connectivity [1]. The cortex is also composed of cortical columns, that show denser interlaminar synaptic connectivity—within a column—than horizontally among adjacent columns [2]. Ensemble neuronal activity within each cortical column generates extracellular currents, which can be measured using microelectrodes with relatively high spatial precision; or more coarsely as net average currents through local field potential recordings [3]. Please see glossary list, if required, for technical terms in the paper (S2 Text).

There are numerous approaches to modelling the dynamics of neuronal activity at the scale of cortical columns. One-dimensional models describing simplified single-neuron dynamics with threshold firing; namely, *integrate-and-fire neurons* [4] can model the impact of interneuron connectivity on larger scale dynamics, such as those observed within and between cortical columns. However, these equations often result in integral-differential equations or partial differential equations (please see S2 Text for definition), which are usually difficult to solve, at least in closed form, and especially when considering larger neuronal systems [5,6].

Using approximation methods from statistical physics, the collective activity (or ensemble activity) of cortical microcircuits comprising populations of distinct cell types have been modelled as neural mass point processes (please see S2 Text for definitions) [7–12]. There are broadly two sets of neural mass models: (1) Conduction-based models based on the Hodgkin-Huxley model, with specific modelling of the intrinsic dynamics of ion channels [13], which can allow for multi-compartment extensions [14,15]; and (2) convolution-based models where simple synaptic kernels are used to estimate the postsynaptic depolarization from the presynaptic input. These convolution-based models are computationally efficient whilst allowing a range of complex dynamics to be simulated. The canonical microcircuit model (CMC) is a typical example of these convolution-based neural mass models. It comprises 4 populations and has been used extensively to model EEG and MEG data [16]. In this setting, an idealized cortical column is modelled using a set of $2^{nd}$-order differential equations with nonlinear coupling, using a sigmoid function to map the effect of the neuronal potential of one population on the depolarization of another (please see S2 Text for definitions).

In this study, we present an analysis of the dynamics of convolution-based neural mass models, specifically, the canonical microcircuit model [16]. When considering the spatially coarse-grained dynamics—typically recorded in human neurophysiological recordings—these kinds of models afford a balance between accuracy and complexity. In other words, they are sufficiently expressive to capture neuronal dynamics, at the scale of a cortical column, while being sufficiently simple to preclude overfitting, when used to explain empirical timeseries.

Electrical activity of the cortex can be readily recorded using electro- or magnetoencephalography (EEG/MEG). Visual analysis of the ensuing timeseries reveals a mixture of apparently stochastic features, such as paroxysmal discharges, intermixed with recurrent rhythms. The

spectrum of rhythms and paroxysmal discharges vary in frequency, location and progression and are seen in a variety of states of the normal and diseased brain [3]. Quantitative analysis of brain states suggest that these recurrent, recognizable patterns can be modelled as semi-stable states, i.e., states with local but not global stability [10,17–20]. Importantly, these dynamical features are sensitive markers for whole-brain (dys)function and can indicate pharmacological and pathophysiological changes of neuronal connectivity at the synaptic level [21–23]. Thus, developing biophysical models of cortical dynamics may help identify the synaptic and connectivity processes that shape paroxysmal and time-varying dynamic features of brain activity.

The dynamical structure of these systems—such as neural mass models—can be characterized using a phase space representation (please see S2 Text for definition). Simulation of the dynamics of neural mass models of the cortical column have disclosed complex structures in this phase space [24–26]. These include stationary points, limit cycles and chaotic attractors (please see S2 Text for definitions). Yet many popular current approaches of fitting neural mass models to empirical data, such as dynamic causal modelling, assume that the dynamics exist in a quasi-steady state, without transitions between distinct dynamical modes [27]. Thus, there is a need to develop the tools necessary to link biophysical models of cortical function to more complex, time-varying, and paroxysmal brain dynamics (e.g., seizure activity in epilepsy).

The theoretical and empirical applications of neural mass models—with separation of time scales—to analyze electrophysiological data, especially seizure activity, are diverse [28–33]. A seminal paper, by Jirsa et al., elegantly modelled epileptic seizure activity, in terms of bifurcations, using a neural mass model with two neural masses and a "permittivity" variable which introduced a separation of time scales. The slow permittivity variable reflects chemical changes, such as variations in electrolyte or metabolic homeostasis. This work has further been elaborated in computational pipelines that can be applied to specific patient data recorded in the evaluation for epilepsy surgery [34]. The Epileptor model differs from the canonical micro-circuit model, as the latter does not have explicit multiple time scales. However, we show in this paper that neural mass models do have an implicit time scale variation, which could explain the phenomena described by the Epileptor model. The method we use to study the implicit time scales of neural mass models are standard techniques in the study of dynamical systems [35,36] and was alluded to by the authors of the Epileptor model [37].

In what follows, we will estimate the semi-stable (i.e., multi-stable) states of the canonical microcircuit (CMC) model. We use an adiabatic approximation to integrate over fast changing variables—to derive the implicit slow time scale dynamics. To this slowly evolving state we add Brownian noise (to model unknown variations of the model, see S2 Text for definition) allowing for transitions between semi-stable states. More specifically, the dynamics of mean activity become gradient flows on a potential function (please see S2 Text for definitions). Crucially, this formulation can be used to generate key measures such as the frequency and duration of transient oscillatory modes, ie. features of itinerant activity. Using this formulation, we briefly consider variational methods that could be used to estimate synaptic connectivity from empirical measures of occupancy and mean exit times from distinct modes of activity. The theoretical results speak to a potential characterization of dynamical itinerancy seen in disorders of the brain—including epilepsy—in computationally tractable ways that can be incorporated in established (variational) model inversion schemes, such as dynamic causal modelling.

## Methods

This section describes the formal results upon which the proposed analysis rests. This analysis can be summarised, in narrative form, along the following lines:

- First, identify a sufficiently expressive neural mass model, whose equations of motion are parameterised in terms of synaptic connectivity.

- With a suitable transformation of variables, separate the fast (oscillatory) dynamics from slow (amplitude) fluctuations, in the spirit of an adiabatic approximation.

- Formulate the dynamics of the slow variables in a relatively simple mathematical form. More exactly, define the dynamics as a gradient flow on a potential function. This potential function can now be taken as the negative logarithm of a steady-state density; namely, the solution of the Fokker Planck equation describing density dynamics under random fluctuations (please see S2 Text for definitions).

- From the steady-state density, evaluate the probability of occupying various fixed points—in the state space of the slow variables—to generate the statistics of dwell times and transitions.

- Use the above steps as a generative model—that generates the statistics of dwell times from synaptic connectivity—to infer connectivity from empirically observed statistics of (slow) transitions among (fast) oscillatory modes of activity.

In what follows, we describe the mathematical basis of the steps above and present a brief, illustrative application, using simulated data to recover the connectivity giving rise to transient bursts of oscillatory activity.

## CMC-model dynamics

The electrical activity of the cortical column can be modelled using the CMC-model as described in [16,27]. To keep the derivations simple, and to ensure readability, we will rename some parameters here, without changing the dynamics of the system. The model considers four distinct neuronal populations modelled using a set of four 2nd order differential equations. These ordinary differential equations can be written equivalently as eight first order equations.

$$\dot{p}_i = -\omega_i^2 q_i + \varepsilon \omega_i^2 \sum g_{ij} S(q_j) + \mu \omega_i^2 \sum g_{ij} P\left(\frac{p_j}{\omega_j}\right)$$

$$\dot{q}_i = p_i$$

Where the potential of the *i-th* neural population is given by $q_i$ and the rate of change of this is given by $p_i$ (i.e. a current term). We have introduced 2 variables $\mu$ and $\varepsilon$ to control the effect of perturbation of the non-connected model where the connectivity matrix is given by $g_{ij}$. A sigmoid function (*S*) is used to parametrize the interaction between the potential and current terms of the neuronal populations. The *P*-function parametrizes spike rate variability.

The original variables ($q_i$, $p_i$) can be written as a complex variable ($\in \mathbb{C}$) which will keep the derivations relatively concise and easy to follow.

$$z_i = q_i + i\frac{p_i}{\omega_i}$$

The closed loops of the unperturbed motion will be given by,

$$z_i z_i^* = R^2$$

The equation of unperturbed motion will be given by,

$$\dot{z}_i = -i\omega_i z_i$$

The solution will be given by,

$$z_i = Re^{-i\omega_i t}$$

The motion of the trajectories can then be given by the modulus ($R$) and argument ($\theta_i$) of the complex number ($z_i$), Fig 1.

We introduce a new phase variable $\varphi_i$ describing the deviation of the phase flow around its value when $\mu$ and $\varepsilon = 0$, i.e. no perturbation.

$$\theta_i = -\omega_i t + \varphi_i$$

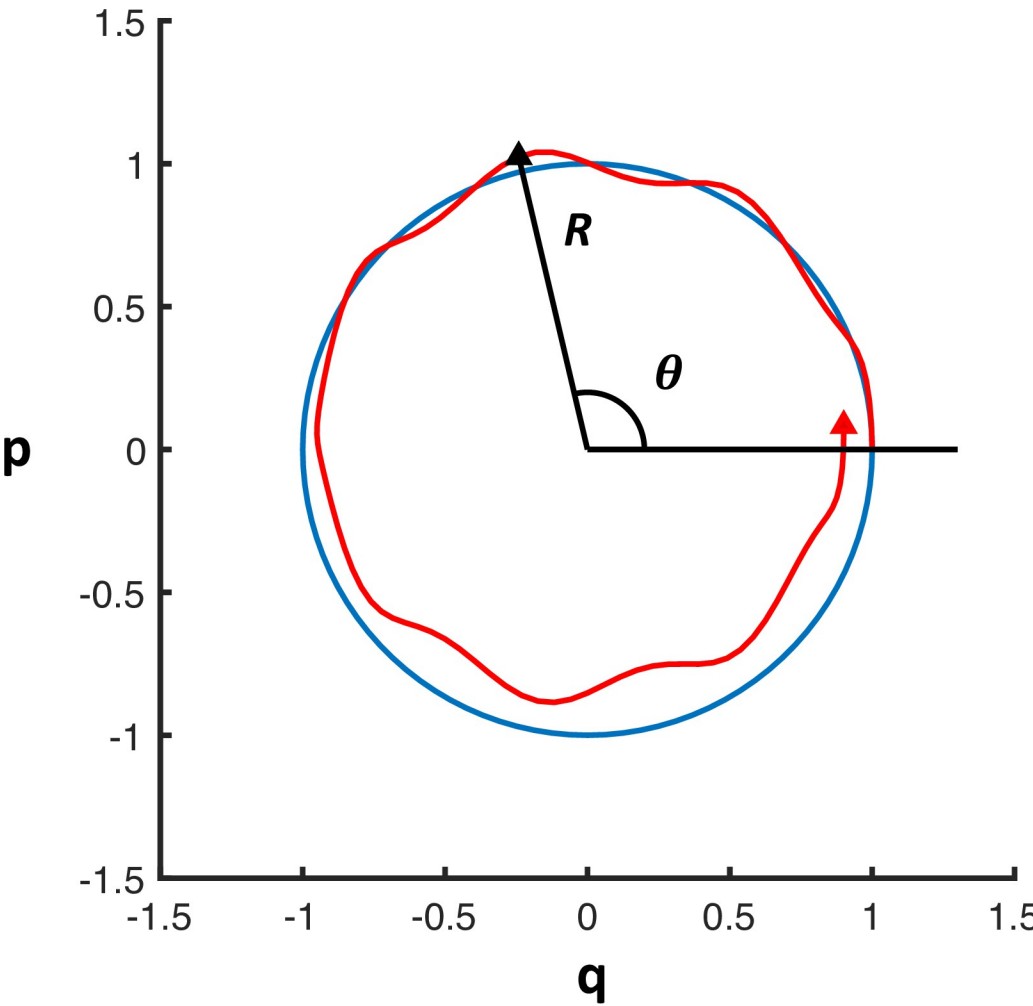

**Fig 1. Schematic figure showing the relation between the variables (q,p) and the amplitude and angle variables.** The blue curve shows the trajectory of an unperturbed trajectory (no damping and no coupling to other neuronal populations). The red curve depicts the trajectory of a perturbed path. The amplitude ($R$) and angle variable ($\theta$) for a point of the new trajectory (red) is shown.

The resulting equations of motion can then be given by,

$$\dot{z}_i = -i\omega_i z_i + i\varepsilon\omega_i \sum g_{ij} S\left(\frac{z_j + z_j^*}{2}\right) + i\mu\omega_i \sum g_{ij} P\left(\frac{z_j - z_j^*}{2i}\right) \tag{1}$$

## Adiabatic approximation

The above equations of motion can be simplified under the assumption that the time scale for phase dynamics (which is determined by $\omega_i$) is small in relation to the time scale for amplitude dynamics. This assumption has empirical validity when modelling cortical activity, as the amplitude modulation of cortical rhythms or spike amplitude evolves over a slower time scale (at least 10–100 times slower) than that of the phase dynamics [28,38].

In the above formulation this adiabatic assumption can be satisfied by limiting the range of $\varepsilon$ and $\mu$. The resulting adiabatic approximation is a simplified model of the amplitude variables (i.e., a 4-dimensional model), which permits further analysis of the dynamics in terms of closed form solutions. The usefulness of the ensuing adiabatic approximation can be assessed by the ability of the model to explain empirical data (see final section).

An adiabatic approximation can be specified with a time duration ($T$), where each neuronal population will complete—to $0^{\text{th}}$ order (see *Perturbation analysis*)—a series of cycles. The angle variables move on a 4-dimensional torus (please see S2 Text for definition), and $T$ is chosen such that the motion over the torus completes (or approximately completes) one cycle and the angle variables can be eliminated from the dynamical flow [35].

$$T = least\ common\ denominator\ of\ \{\omega_i\}_{i=1,..,4}$$

## Results

### Further analysis of the dynamics

A perturbation analysis of Eq 1 can now be pursued using the following expansion for $\varphi_i$ and $R_i$ (in modulus-argument form, please see S2 Text for definition).

$$z_i(t) = R_i(t,\varepsilon,\mu)e^{-i(\omega_i t + \varphi_i(t,\varepsilon,\mu))}$$

Where the two terms will be given by,

$$R_i(t,\varepsilon,\mu) = R_{i,0,0} + \sum \varepsilon^m \mu^n R_{i,m,n}(t)$$

$$\varphi_i(t,\varepsilon,\mu) = \sum \varepsilon^m \mu^n \varphi_{i,m,n}(t)$$

From the flow of states, we get the following,

$$\dot{z}_i(t) = \dot{R}_i e^{-i(\omega_i t + \varphi_i)} - iR_i(\omega_i + \dot{\varphi}_i)e^{-i(\omega_i t + \varphi_i)}$$

Which can be written as,

$$\left(\dot{R}_i(t,\varepsilon,\mu)e^{-i(\omega_i t + \varphi_i)} - iR_i(t,\varepsilon,\mu)\dot{\varphi}_i e^{-i(\omega_i t + \varphi_i)}\right) = i\varepsilon\omega_i \sum g_{ij} S\left(\frac{z_j + z_j^*}{2}\right) + i\mu\omega_i^2 \sum g_{ij} P\left(\frac{z_j - z_j^*}{2i}\right)$$

The resulting equation can now be integrated over the closed (or almost closed) path over the torus, such that the angle variables are eliminated. To perform this integration we expand the perturbative functions S and P as polynomial series. We are now in a position to derive the

change in $R_i$ to 1st order, by combining the results derived in the Appendix (S.1.1–3):

$$\frac{dR_i}{dt} = \frac{\delta R_i}{T} = \frac{\mu R_{i,0,1}(T)}{T} = \mu \omega_i g_{ii} \sum_{r=1}^{\infty} B_r \frac{R_{i,0}^{2r-1}}{2^{2r-1}} \binom{2r-1}{r-1}$$

The dynamics of $R_i$ is determined by 2 coupling terms: the potential-to-current coupling ($S$) and the current-to-current coupling ($P$). The nonlinearity of the coupling functions (S and P) result in a series expansion in multiples of $R_i$. The complexity of the dynamics for the amplitude parameters is determined by the connectivity matrix and the coupling functions.

These parameters are defined in the cortical column by reference to the synaptic connections between the neuronal populations, where the gain of each connection is given by $g_{ij}$ and their physiological effect on the target neuron parametrized in S and P. Note that the above derivation is valid for any set equal to or larger than two neuronal populations.

## Dynamics of amplitude modulations

The electrical or current recordings of cortical activity is often modelled as a linear combination of the activity of the potential of each neuronal population [39]. This can be used to derive a 1-dimensional equation of motion for the amplitude dynamics of the recorded data. As can be seen above the amplitude dynamics will be governed by the self-connection terms, $g_{ii}$. The self-interactions within the cortical column will result in a set of non-interacting limit cycles (taking up-to 1st order contribution from the perturbation).

## Amplitude envelope dynamics

The measured potential (e.g., EEG) is given by (assuming a linear lead field).

$$y(t) = \sum a_i R_i \cos\omega_i t \tag{2}$$

The amplitude power can be estimated easily, provided the $\omega_i$ are distinct.

$$R^2 = \sum a_i^2 R_i^2$$

We can now normalize $q_i$ s.t

$$\sum_{i=1}^{n} a_i^2 = n$$

$R$ will then be the radius of a sphere if all $a_i$ are constant in the vector space given by $R_i$. The rate of change of the amplitude function $R$ is given as below.

$$R\frac{dR}{dt} = \sum a_i^2 R_i \frac{dR_i}{dt}$$

As we are interested in the average rate of change of $R$ we will take a spatial average over ellipsoids with constant $R^2$.

$$\frac{1}{A} \oiint R\frac{dR}{dt} = \frac{1}{A} \oiint \sum a_i^2 R_i \frac{dR_i}{dt}$$

$R$ will be constant on the surface,

$$\frac{1}{A} \oiint \frac{dR}{dt} = \frac{1}{AR} \oiint \sum a_i^2 R_i \frac{dR_i}{dt}$$

Hyperspheroidal coordinates (please see S2 Text for definition) are used for the integration (see S1 Text, S1.4). Let $R_s$ be the average of $R$ over the 3-sphere,

$$\frac{dR_s}{dt} = \frac{1}{AR}\oiint\sum a_i{}^2 R_i \frac{dR_i}{dt} = \mu\sum \omega igii\sum_{r=1}^{\infty} B_r \frac{4R^{2r-1}}{(2r+1)2^{2r}a_i{}^{2r}}\binom{2r-1}{r-1}\frac{(2r+1)!!}{(2r+2)!!} \qquad (3)$$

As was discussed in the appendix, cross connectivity terms (or off-diagonal terms) will affect the dynamics of $R_s$ when higher order terms are included in the perturbation expansion. The functional form of this equation means that the amplitude dynamics can be expressed as a gradient flow on a potential, $U$ that depends upon the synaptic connections $g_{ij}$ and the coupling functions:

$$\partial_t I = -\nabla U$$

In other words, different connectivity matrixes and functions correspond to different potentials ($U$): see Fig 2. This spectrum then defines different possibilities of dynamics supported by the CMC model. Panel A shows minima at 0 and 2. Panel B has three stable points, at 0, 1 and 2 (see S1 Text, S1.1–3). This suggests that the CMC model—with certain synaptic connections—features continuous high amplitude oscillations and a point of stability at the

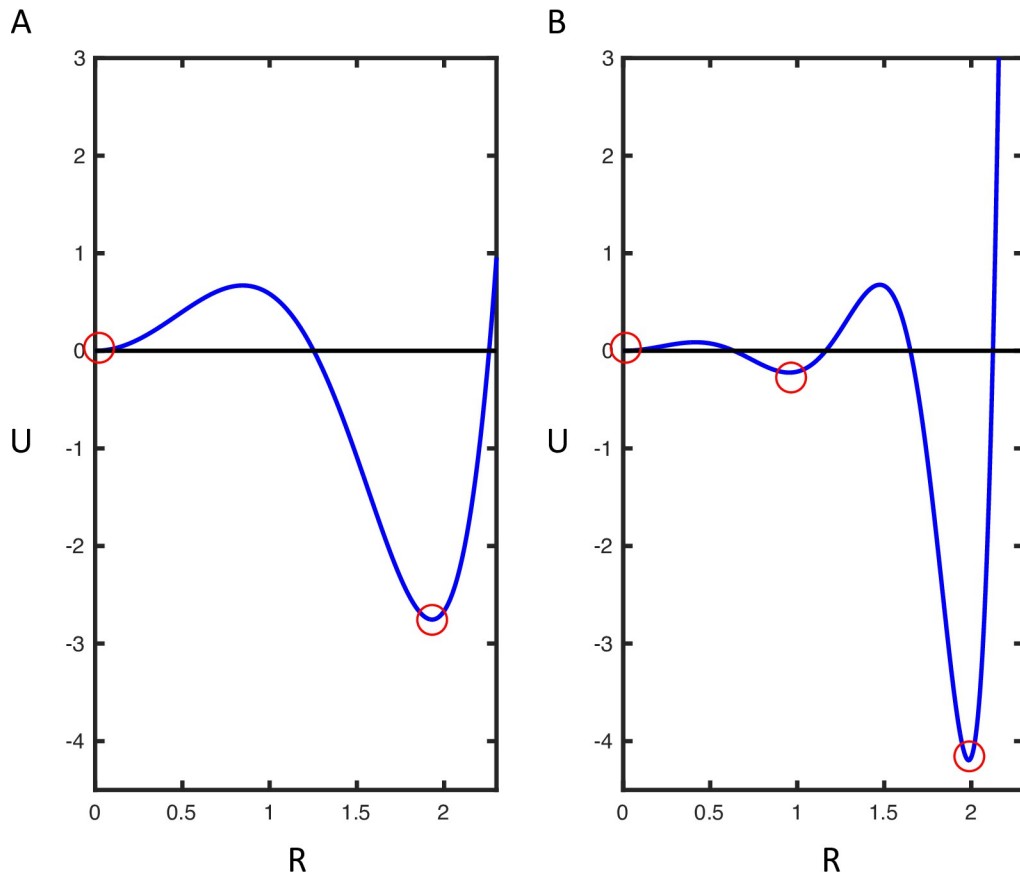

**Fig 2. Potential distributions for the dynamics of the total amplitude ($R$) for different connectivity between the neuronal subpopulations.** Here different connectivity matrices and couplings are selected to illustrate the spectrum of potential functions. Minima of $U$ in these functions are stable point attractors (A) The first example has two stable points at the origin and at R = 2. (B) The second figure has three stable states at the origin and at R = 1 and 2.

origin. The oscillations would be a mixture of sinusoidal activity, with frequencies of the four neural populations, where this mixture is determined by the lead field (Eq 2).

The potential distributions $U$ in Fig 2 are generated by one self-connectivity term in the superficial pyramidal cell layer. Up until this point, we have been considering deterministic dynamics in the absence of noise. In what follows, we show that the same dynamical structures persist in the presence of random fluctuations, via an analysis of the density dynamics.

A schematic of the steps taken to derive the dynamical equations of the amplitude modulation of the EEG is shown in Fig 3.

## Stationary states and exit times

Fast neuronal fluctuations can be easily modelled by adding innovations to the deterministic equations of motion [40,41]. For the potential flow of the amplitude dynamics derived above we can add a white (Wiener) noise process, resulting in a stochastic differential equation (SDE) (see S2 Text, for definitions).

The trajectories are then described by the following SDE,

$$dI = -\nabla U dt + \sigma dB_t$$

In the above equation, $I$ and $B_t$ are random variables while $U$ and $\sigma$ are real valued functions. The associated Fokker Planck equation, describing the density dynamics over the amplitude, is given by:

$$\partial_t \rho = \frac{\sigma^2}{2} \Delta \rho - \nabla.(\nabla U \rho)$$

The stationary state of this equation is given by the following, where $Z$ is the partition function (please see S2 Text for definition) or equivalently, a normalizing constant for $p$).

$$p_{st} = \frac{1}{Z} exp\left\{-\frac{2U}{\sigma^2}\right\}$$

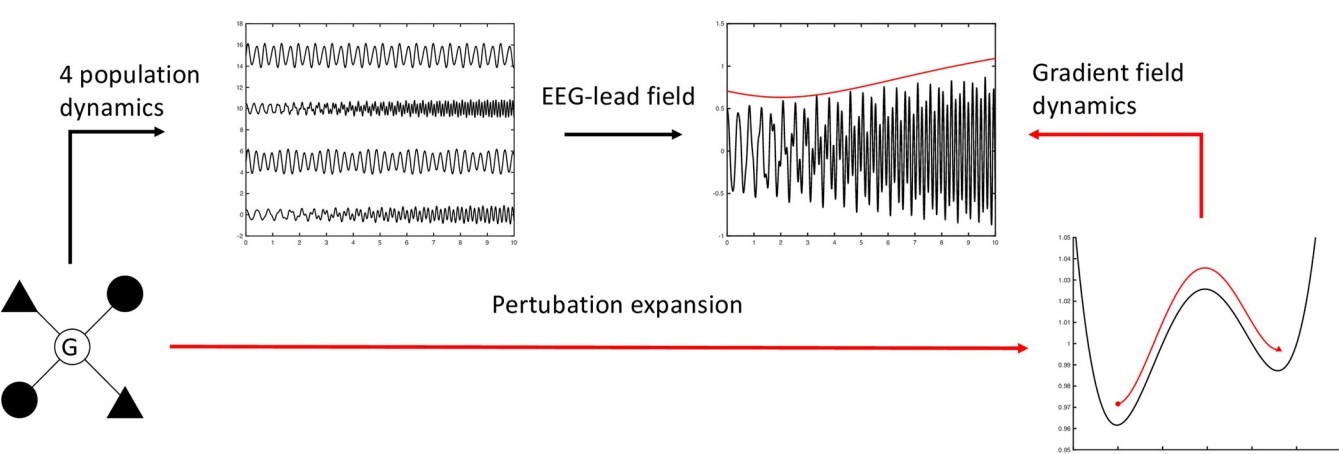

**Fig 3. A schematic figure of the steps taken to derive the amplitude dynamics.** The starting point is the 4 population CMC-model. The activity of the model generates activity for each of the four populations as shown after the arrow "4-population dynamics". The EEG lead field will then take a linear mixture of the activity of the 4 populations resulting in a "EEG" curve. The amplitude modulation of the activity is drawn by a red curve. The adiabatic and perturbation analysis discussed in this paper will define a gradient field as drawn after the arrow "perturbation expansion". The gradient field will depend on the connectivity matrix, G. A possible trajectory of the amplitude variable is shown as a red arrow moving from one stable state to another. The corresponding change in amplitude over time is shown after the arrow "gradient field dynamics" in red above the "EEG curve". The main contribution of this work is to find a direct relation between the intrinsic connectivity and the gradient field which in turn will define statistical measures such as the probability of being in a specific state or exit times from states.

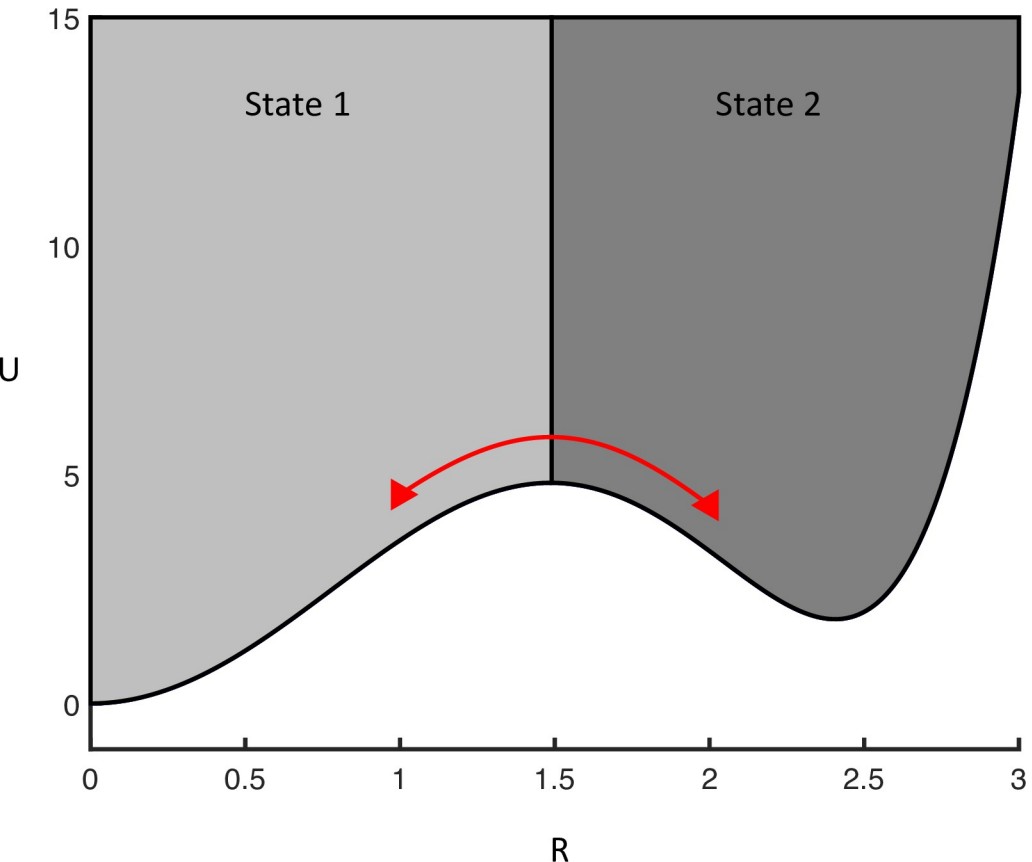

**Fig 4. Regions attracted to different minima of the potential (shaded differently).** The probability of finding the system in one of these stable states is given by integrating the probability density function over the attracting interval (state 1 = light grey), 0 to the first maxima ($R_{max}$) of the potential curve or $R_{max}$ to $\infty$ (state 2 = dark grey). The red arrow depicts how the trajectories pass between state 1 and 2 due to the underlying noise.

With a connectivity matrix that features several stable states, we can now estimate the probability of finding the dynamical system in that state by simply integrating the stationary probability density function (i.e., solution to the Fokker Planck equation above) over the interval defining the state: see Fig 4. This provides an analytic estimate of noise induced multistability (c.f., topological supersymmetry breaking), in terms of the probability of finding the amplitude dynamics near one of the fixed points or stable (oscillatory) states.

For the first state the probability is given by,

$$P\{state\ 1\} = \int_0^{R_{max}} \rho_{st} dI$$

The mean duration of time ($t$) the system is in a given state can be approximated using Kramer's escape rate,

$$\langle t \rangle = \frac{1}{2\pi \sqrt{-U''(x_{min})U''(x_{max})}} e^{\frac{2[U(x_{max})-U(x_{min})]}{\sigma^2}}$$

Finally, because we have closed form expressions for both the state probability and mean exit times, we are able to estimate the connectivity matrices from empirical measures of occupancy and mean exit times from discernible oscillatory states. In other words, the closed form

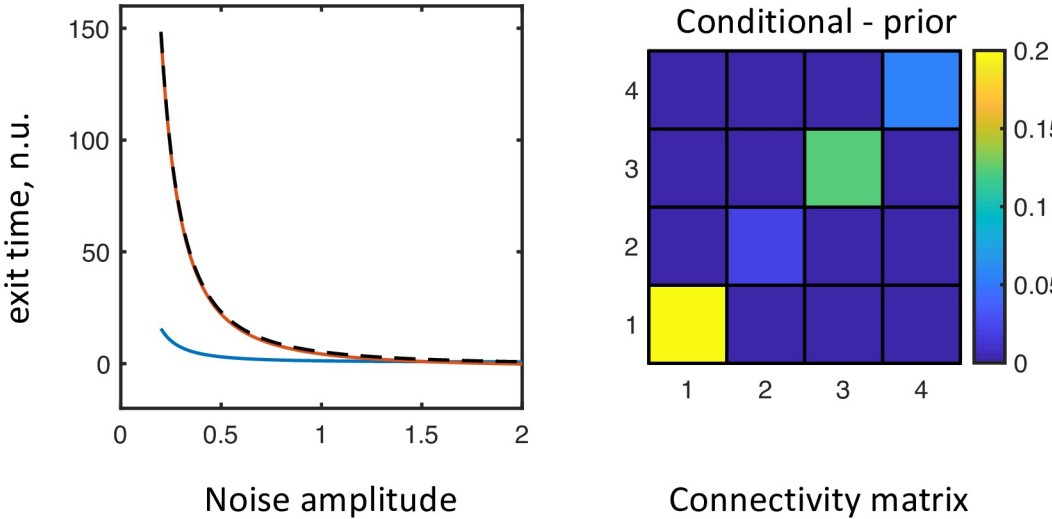

**Fig 5. Inversion of the generative model based on simulated data.** The left panel shows the mean exit time (on the vertical axis) from state 2 (Fig 4) for different noise amplitudes (plotted along the horizontal axis). The dotted black curve indicates the synthetic data that was inverted. The blue curve shows the expected data under the prior values of the connectivity matrix. The red curve indicates the expected exit times with the posterior values of the connectivity matrix. The right panel shows the change in the connectivity matrix between the prior and posterior estimates of the connectivity matrix, suggesting that the connectivity parameters used to generate the data can be recovered from mean exit time data features.

expressions above provide the basis for a generative model that can be inverted given empirical measures of transitions among oscillatory modes evident in any empirical data. See Fig 5 for inversion of simulated data of exit times, for different noise amplitudes, using the variational procedures used in dynamic causal modelling. A simulated EEG trace is shown in Fig 6 illustrating how the system is moving between two semi-stable states resulting in a bursting appearance of the EEG.

In this proof of principle example, synthetic data were generated from using prior values of the connectivity in the CMC model provided by the SPM software. Inversion of the synthetic data was performed using standard (variational Laplace) inversion routines available in the SPM software package. The code used in this paper can be obtained from https://github.com/gercoo/exittimes2synapticconnectivity. This example shows that the connectivity parameters used to generate transitions among oscillatory states or modes can be recovered from mean exit times alone. The implication here is that it is possible to both explain and leverage multistability in terms of underlying synaptic connections and the amplitude of fast neuronal fluctuations. This work could be applied when investigating spiking in epileptic and non-epileptic tissue as the inter-spike interval could be modelled as the exit time from a large amplitude transient. The inversion method described could then map spike frequency to synaptic connectivity, which could be used to uncover the underlying physiological differences between regions of the brain with different epileptogenicity.

## Discussion

This paper has presented a formulation of multi-stable neuronal dynamics based on neural mass models—and an accompanying analysis framework—that enables inference about synaptic connectivity parameters that engender to observable transitions among oscillatory states. We have derived closed form expressions for the statistics of the semi-stable states, e.g. the state probability and the mean duration of a state.

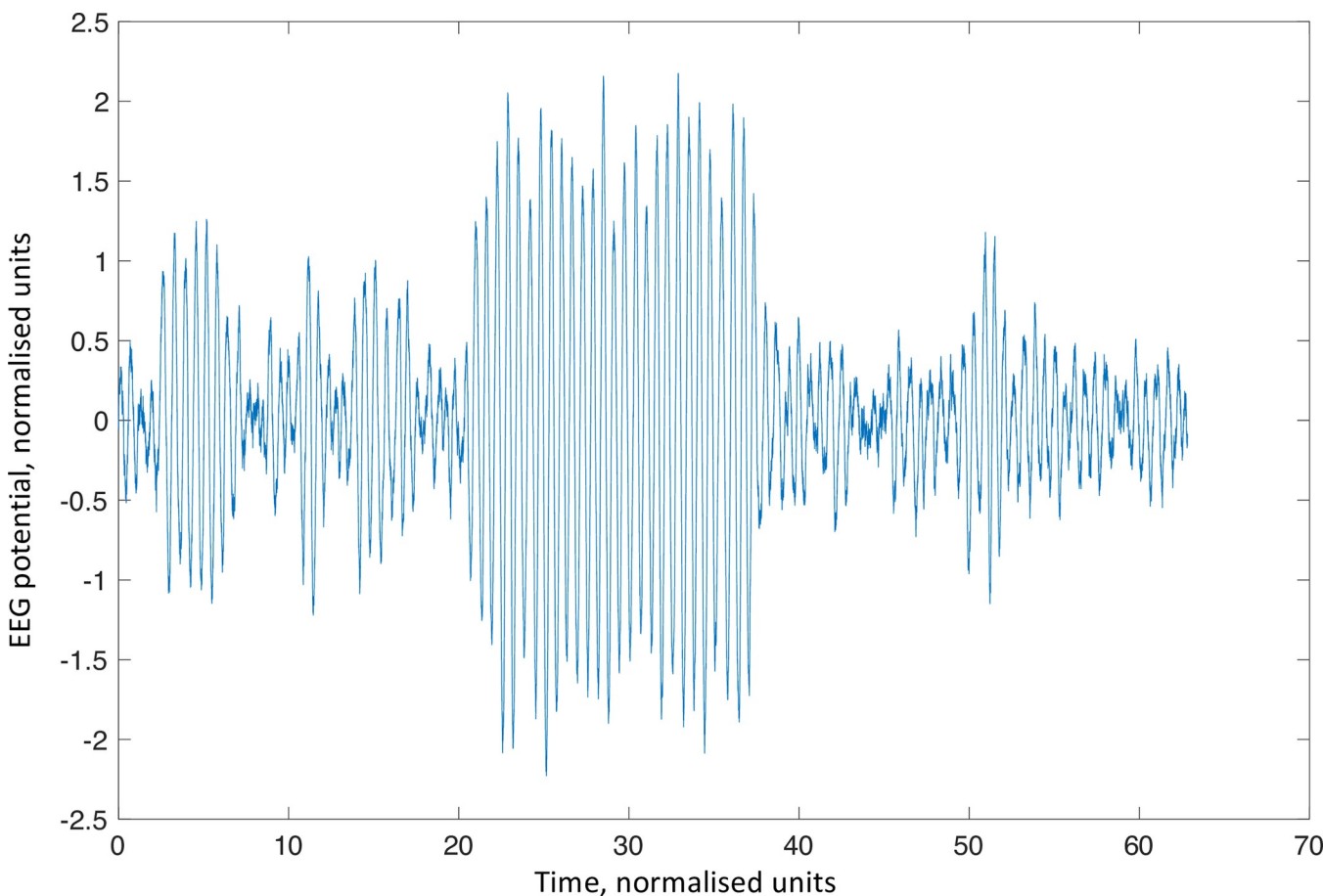

**Fig 6. Simulation of EEG trace with bi-stable activity.** The trace shows a simulated EEG trace with a connectivity matrix giving a bistable system as described in Fig 4. The system is bursting with high amplitude oscillations (state 2) with shorter suppressions of attenuation (state 1) in between.

Our work builds on both theoretical insights of prior neural mass modelling and empirical observations of transient brain states. Prior simulations of neural mass models have already shown the simultaneous presence of fixed point and limit cycle dynamics in the same cortical model [25,42]. Furthermore, empirically there are several instances—in both the healthy and diseased brain—where the mode of cortical and subcortical dynamics shifts repeatedly among states; including physiological rhythms like sleep spindles, mu-rhythms, and pathological states, such as epileptic seizure activity [3,43–47]. Indeed, the implicit itinerancy may be the hallmark of all self-organising dynamical systems; ranging from the stochastic chaos that characterises nonequilibrium steady-state [48–55] through to heteroclinic cycles [56–58].

The neural mass models we have considered here show complex dynamics that can be divided into explicit fast and implicit slow variables. This separation in time scales allows for an adiabatic approximation, which eliminates the fast modes producing a new set of dynamical equations of slow variables. This new set of slow variables now describes slowly evolving amplitude dynamics, rather than the fast oscillatory activity itself. This reduces the dimensionality of the system making it more tractable for simulation and inference. In this reduced system, several types of fixed points—representing stationary states or oscillatory limit cycles—can be characterised using measurable data features such as dwell and mean exit times.

In this work, we have presented closed form equations that link patterns of cortical microcircuit synaptic connectivity to statistical data features. This generative model allows for

Bayesian model inversion and comparison, allowing empirical comparison of different mechanistic explanations of intermittent cortical dynamics. This might be particularly interesting when modelling relatively fast changing dynamics like on and off rhythmic spiking or seizure onset. As seizures progress there are changes in several variables including electrolytic or metabolic changes which would have a large but secondary effect on cortical dynamics and might change some key connectivity parameters, which would alter both fast and slow dynamics in the system described here [59]. These slower changes can also be modelled effectively using *Adiabatic DCM* where the effect of slow changes in synaptic parameters is included in the model inversion [60].

The amplitude dynamics of a single-source CMC model (i.e., cortical column) has been shown to be sufficiently approximated by a second order perturbation expansion. In contrast, extrinsic connections between cortical columns may require higher order perturbation terms and thus be of weaker strength compared to the intrinsic dynamics (at least within the parameter range where the perturbation expansion is valid). This is due to the number of neurons required to create an interconnected loop between two cortical columns [2,61]. The intrinsic (i.e., within-source or column) dynamics is thus instantiated in cortical columns through the intrinsic connectivity matrix, while the extrinsic connectivity results in a weak but richer set of dynamics, as the extrinsic (i.e., between-source or column) connectivity can be diverse. This is in line with the architectural setup of cortical processing, where cortical columns represent the smallest units that process data as conditionally independent units and interact with other columns, using a weaker set of interactions [62]. We will further develop the perturbation theory presented here to include higher order terms to see whether the interactions between cortical columns can be characterised.

Our perturbation expansion is sufficient to explain semi-stable states of the cortical modes of activity (i.e. stationary states and limit cycles of the full dynamics) within cortical columns in the absence of changes of connectivity. Using a perturbation expansion, we have managed to derive the characteristics of semi-stable states of the cortical columns from the connectivity matrix describing connectivity within the cortical column. This is an important step, as we can now map statistical characterisations of complex cortical dynamics onto synaptic connectivity. Understanding larger cortical networks from a modelling perspective can be further implemented using a discrete simplification of the above model. It is possible—through a projection mapping of the dynamics of the cortical columns onto the semi-stable state it occupies at any given time—to simplify things considerably. This should preserve the important link between connectivity and the semi-stable states that are generated. Connecting different cortical columns could then result in a lattice model (similar to an Ising or Pott's model) of the cortex, where it would be relatively straightforward to link the intrinsic and extrinsic connectivity to measurable dynamics [63,64]. Such an expansion of our work would prove particularly relevant to the characterisation of epileptogenic networks and their role in seizure initiation and spread. Furthermore, the effect of medication could be addressed in terms of the changes in the intrinsic connectivity and in ceiling changes in network dynamics. Finally, the effect of epilepsy surgery or disconnection surgery could be modelled by changes in extrinsic connectivity; c.f. [65–67].

It is important to compare this work with other published methods using neural mass models to investigate itinerant brain dynamics. The Epileptor model is frequently used to characterize epileptic seizure onset, propagation and offset [28,34,66]. As described previously, the timescales are explicit in the functional form of the equations of the Epileptor model; while they are only implicit in the physiologically grounded (canonical microcircuit) model described in this study. The explicit parameterization in terms of distinct rate (i.e., time) constants simplifies the mathematical modelling of seizures and provides a compelling taxonomy

of seizures in terms of bifurcations for the field of computational epilepsy. The CMC-model, in contrast, requires extensive mathematical derivations to uncover the underlying slow processes; however, the biophysics of slow modulation can be shown to depend on the synaptic connectivity within a cortical column. This may be important in terms of understanding the mechanisms (e.g., synaptopathies) that underwrite epilepsy.

We have thus shown the possibility of a cortical column evincing slow dynamics in the absence of slowly fluctuating variables. This dynamical mechanism fits well with the principle of self-organization, where a system shows patterns in time or spatial scale larger than the local dynamics governing the system. The analysis provided in this paper further allows for the statistics of transient electrophysiological activity (e.g., spike frequency and seizure duration) to be mapped onto the synaptic connectivity, in contrast—or as a complement—to the permittivity variable in the Epileptor model.

Finally, there is a key difference between accounts exemplified by the Epileptor model, and the model described in this paper. The dynamics of slow variables (e.g. amplitude modulation) derived in this paper are dependent on fast variables (e.g. high frequency activity within neural populations) only via the adiabatic approximation. In other words, the fast activity pushes the system configuration slowly giving rise to the implicit slow dynamics. We describe a causal flow from fast to slow timescales, in contrast to a feed-back system of the kind found in the Epileptor model, where the dynamics of the slow variables directly affect faster variables. The dependence of slow variables on fast variables in this paper allows us to define an "arrow of causality" that is similar to that found in physics, where macroscopic variables can be estimated from the underlying microscopic variables but not vice versa. [65,68–71]. However, we note that the feed-back from slow to fast timescales may be required in certain cases (e.g., periodic seizures or seizures of longer duration), perhaps using a similar set-up as the Epileptor model [28]. This speaks to some interesting issues about the nature of circular causality, and bottom-up versus top-down causation [66, 72]. These issues could, in principle, be addressed using Bayesian model comparison of appropriate dynamic (causal) models.

The approach illustrated here may have relevance for our understanding of paroxysmal disorders of cortical function such as epilepsy and epileptic seizures. The epilepsies are characterised by recurrent, sudden onset of rhythmic epileptic activity, on a background of spontaneous 'normal' interictal ongoing activity. In certain cases, cortical activity in epilepsy can alternate rapidly between more than two states, 1) rhythmic spiking, 2) seizure activity and 3) spontaneous background activity [17,73]. The transition between these states has been described in terms of phase transitions and bifurcations, where the synaptic parameters that shape microcircuit dynamics exceed a threshold, resulting in a sudden change in phase space dynamics [39]. Our model, similar to previous modelling endeavours, offers an alternative perspective in that it does not require a critical transition (as in transcritical bifurcations) to explain the change in cortical activity [74]. Rather, the mechanism resulting in intermittently observable dynamic phenomena is predicated on noise induced itinerancy, of the kind found in stochastic chaos and topological supersymmetry breaking. In other words, the stochastic nature of the model will move the dynamics of the system between semi-stable states. This form of transition between states has been described in terms of transcritical activity [39]. What is new in this study is that we can now map the observed changes to synaptic connectivity within the cortex.

Observations of intermittent dynamic features could—in principle—give support for either bifurcations or transitions between semi-stable states. Note that we do not provide any evidence for the absence of bifurcations in these intermittent systems, as these still exist within the framework presented (i.e. changing the connectivity matrix can still cause transcritical bifurcations): rather that these are not always required to explain observable state transitions. To distinguish between the two proposals; bifurcations or semi-stable state transitions, the

statistics of transitions among states may need to be further explored. It is through tools such as the invertible, reduced models presented here that future work will be able to specifically address these questions empirically; i.e., through the (Bayesian) model comparison of models with and without changes in connectivity.

In summary, we have introduced the theoretical backdrop for a model of cortical multi-stability and itinerancy that could be used to infer the underlying synaptic connectivity from statistical descriptions of transient dynamic features. We hope to elaborate the theoretical work presented above for the characterisation of global dynamics based on measurable electromagnetic responses in health and disease.

## Supporting information

**S1 Text. Appendix with in-depth mathematical analysis.**
(DOCX)

**S2 Text. Glossary list with technical terms used in the paper.**
(DOCX)

**S1 Fig. 2D-ellipoids representing the surfaces with constant EEG amplitude.** The spatial average of the amplitude flow will be done over the positive sector ($Ri>0$) of each of these surfaces.
(TIF)

**S2 Fig. 2D-spherical and 2D-ellipsoidal area unit.** The spherical (yellow) and the ellipsoidal (blue) area unit are separated for better visualization. Using the spherical area unit instead of the elliptic will simplify the integrals considerably allowing for analytical solutions.
(TIF)

## Author Contributions

**Conceptualization:** Gerald Kaushallye Cooray, Karl John Friston.

**Formal analysis:** Gerald Kaushallye Cooray, Karl John Friston.

**Funding acquisition:** Richard Ewald Rosch, Karl John Friston.

**Investigation:** Gerald Kaushallye Cooray.

**Methodology:** Gerald Kaushallye Cooray, Karl John Friston.

**Resources:** Richard Ewald Rosch, Karl John Friston.

**Software:** Gerald Kaushallye Cooray.

**Supervision:** Karl John Friston.

**Validation:** Gerald Kaushallye Cooray, Richard Ewald Rosch, Karl John Friston.

**Visualization:** Gerald Kaushallye Cooray, Richard Ewald Rosch, Karl John Friston.

**Writing – original draft:** Gerald Kaushallye Cooray, Richard Ewald Rosch, Karl John Friston.

**Writing – review & editing:** Gerald Kaushallye Cooray, Richard Ewald Rosch, Karl John Friston.

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
