## [Decision Letter · Decision Letter 0]

5 Dec 2022

Dear Dr. Cooray,

Thank you very much for submitting your manuscript "Global dynamics of neural mass models" for consideration at PLOS Computational Biology.

As with all papers reviewed by the journal, your manuscript was reviewed by members of the editorial board and by several independent reviewers. In light of the reviews (below this email), we would like to invite the resubmission of a significantly-revised version that takes into account the reviewers' comments.

We cannot make any decision about publication until we have seen the revised manuscript and your response to the reviewers' comments. Your revised manuscript is also likely to be sent to reviewers for further evaluation.

Sincerely,

Michael Breakspear

Guest Editor

PLOS Computational Biology

Lucy Houghton

Staff

PLOS Computational Biology

Reviewer's Responses to Questions

**Comments to the Authors:**

Reviewer #1: In their paper, Cooray et al. made a thorough analysis of a popular and effective type of brain dynamics models, i.e. neural mass models, which are a coarse-grained version of neural field models that has shown its relevance in many applications (e.g., epilepsy). The paper has the merit of proposing a detailed mathematical analysis that describes the behavior of those models not only around the fixed points, which is useful to understand the dynamical repertoire of those models.

However, in its current form the paper has some major drawbacks that are listed below:

1) The paper is difficult to follow for a non-specialist. The paper makes an extensive use of technical terms borrowed from different sub-fields of physics (dynamical itinerancy, adiabatic approximation, potential function, equations of motion...) which complexifies reading in an unnecessary manner. Extensive editing should improve this aspect to make the paper more accessible and increase its impact. Technical terms such as those, when unavoidable, should be defined operationally to facilitate reading of the paper.

2) The potential applications listed are not sufficiently described, which is not convincing: the authors should insist on why their approach will provide new insights in epilepsy (where much has been done in terms of neural mass modeling / analysis, more on this point below) at the onset of seizures, or for the description of beta bursts in Parkinson's disease. In its current form, this gives the impression that the authors have just listed some applications but without any specific reason or rationale (and there is likely one I am sure, but it is not obvious in the current form of the paper).

3) The authors mention that one innovation of their approach is the decomposition of the classical neural mass model (that they call "Canonical Microcircuit Model" or CMC) in terms of slow/fast sub-systems, and bifurcation analysis. However, slow-fast decomposition of the CMC, notably in the field of epilepsy, has been already performed previously (see for example this recent study uncited by the authors, https://doi.org/10.1371/journal.pcbi.1008430, which performs slow/fast decomposition at the initiation of seizures). Therefore, the innovation brought by the paper is unclear, and should be emphasized to clarify the potential impact of the paper and its interest for a broad readership.

4) It seems that previous literature (such as on the Epileptor model, https://doi.org/10.1093/brain/awu133) should be more discussed or taken into account, given the overlap with the present work.

Overall, my recommendation would be to request at least a major revision of the paper, along the lines suggested here, to increase its readability, clarify its original contribution and innovation, and accounting more accurately contributions from previous literature and how the present study goes beyond the state-of-the-art. I emphasize again however that the study seems technically sound, and that it has merit.

Reviewer #2: In this paper, the authors present a novel, elegant, and useful way to reduce the complexity of neural mass models of local cortical microcircuits by means of time scale separation, thus making them more accessible to model inversion, especially in larger portions of the state space away from fixed points. Although, personally, I would have liked a somewhat more elaborate evaluation of the methods using simulations and real data, I think the paper is worth publishing as it stands. The paper is very well written, so I have no further comments (which is a very rare case for me).

**Have the authors made all data and (if applicable) computational code underlying the findings in their manuscript fully available?**

Reviewer #1: **No: **Only general websites are listed, the information provided to access the code / data is not sufficiently specific.

Reviewer #2: Yes

PLOS authors have the option to publish the peer review history of their article (what does this mean?). If published, this will include your full peer review and any attached files.

Reviewer #1: No

Reviewer #2: **Yes: **Thomas R. Knösche
---

## [Decision Letter · Decision Letter 1]

27 Jan 2023

Dear Dr. Cooray,

Thank you very much for submitting your manuscript "Global dynamics of neural mass models" for consideration at PLOS Computational Biology. As with all papers reviewed by the journal, your manuscript was reviewed by members of the editorial board and by several independent reviewers. The reviewers appreciated the attention to an important topic. Based on the reviews, we are likely to accept this manuscript for publication, providing that you modify the manuscript according to the review recommendations.

Editorial Note: Please make a final read and edit of the paper, ensuring where possible a clear integration of the technical terms in the main text and subheadings with the glossary as per the reviewer's request

Sincerely,

Michael Breakspear

Guest Editor

PLOS Computational Biology

Lyle J. Graham

Section Editor

PLOS Computational Biology

Editorial Note: Please make a final read and edit of the paper, ensuring where possible a clear integration of the technical terms in the main text and subheadings with the glossary as per the reviewer's request

Reviewer's Responses to Questions

**Comments to the Authors:**

Reviewer #1: The authors have provided a revision that partly addresses the comments raised in the first review. The addition of an appendix with a lay definition of technical terms is very much welcome, however within the body of the paper, nothing has changed much in this regards, as is apparent in the "Track changes" version of the document. Therefore, the paper remains filled, including in the Section titles, with terms that are as I mentioned before unnecessarily complicated.

Furthermore, on the topic of Parkinson's disease, what has been added is very vague, and only "bursts" are mentioned: in which region, at rest, during tremor, during a task? Furthermore no reference is provided at all. I would suggest either to suppress Parkinson's disease from the paper, or to add precise and referenced content. The relevance of citing PD would also be an asset. Otherwise, the authors may as well cite almost any disease.

Also, the "arrow of causality" term is one of those terms that is technical for a large portion of the readership, should be defined, and also should be supported clearly by the results. If the authors want to keep such a term in the paper, this would deserve explaining this in more details.

**Have the authors made all data and (if applicable) computational code underlying the findings in their manuscript fully available?**

Reviewer #1: Yes

PLOS authors have the option to publish the peer review history of their article (what does this mean?). If published, this will include your full peer review and any attached files.

Reviewer #1: No

Figure Files:

Data Requirements:

Reproducibility:

References:

---

## [Decision Letter · Decision Letter 2]

3 Feb 2023

Dear Dr. Cooray,

We are pleased to inform you that your manuscript 'Global dynamics of neural mass models' has been provisionally accepted for publication in PLOS Computational Biology.

Best regards,

Michael Breakspear

Guest Editor

PLOS Computational Biology

Lucy Houghton

Staff

PLOS Computational Biology

Reviewer's Responses to Questions

**Comments to the Authors:**

Reviewer #1: All my comments have been addressed in a satisfactory manner.

**Have the authors made all data and (if applicable) computational code underlying the findings in their manuscript fully available?**

Reviewer #1: None

PLOS authors have the option to publish the peer review history of their article (what does this mean?). If published, this will include your full peer review and any attached files.

Reviewer #1: No

---

## [Editor Report · Acceptance letter]

7 Feb 2023

PCOMPBIOL-D-22-01288R2 

Global dynamics of neural mass models

Dear Dr Cooray,

I am pleased to inform you that your manuscript has been formally accepted for publication in PLOS Computational Biology. Your manuscript is now with our production department and you will be notified of the publication date in due course.

With kind regards,

Zsofia Freund
